# Manipulating directional flow in a two-dimensional photonic quantum walk under a synthetic magnetic field

Quan Lin[1], Wei Yi [2,3] ✉ & Peng Xue [1] ✉

Matter transport is a fundamental process in nature. Understanding and manipulating flow in a synthetic media often have rich implications for modern device design. Here we experimentally demonstrate directional transport of photons in a two-dimensional quantum walk, where the light propagation is highly tunable through dissipation and synthetic magnetic flux. The directional flow hereof underlies the emergence of the non-Hermitian skin effect, with its orientation continuously adjustable through the photon-loss parameters. By contrast, the synthetic magnetic flux originates from an engineered geometric phase, which, by inducing localized cyclotron orbits, suppresses the bulk flow through magnetic confinement. We further demonstrate how the directional flow and synthetic flux impact the dynamics of the Floquet topological edge modes along an engineered boundary. Our results exemplify an intriguing strategy for engineering directed light transport, highlighting the interplay of non-Hermiticity and gauge fields in synthetic systems of higher dimensions.

Open systems are ubiquitous in nature, and exhibit rich and complex behaviors unknown to their closed counterparts[1]. The recent progresses in non-Hermitian physics offer fresh insights into open systems from a unique perspective, giving rise to exotic symmetries and new paradigms of topology[2–9]. A much studied non-Hermitian phenomenon of late is the non-Hermitian skin effect (NHSE)[9–32], whereby a macroscopic number of eigenstates become exponentially localized toward the boundaries. The NHSE has significant impact on the band topology[9–11], the spectral symmetry[33–35], and dynamics[36–38]. One of the most salient dynamic signatures of the NHSE is the directional bulk flow[13,38–40], which is closely connected to the global topology of the spectrum on the complex plane[31,32]. Such directional dynamics can have potential applications in topological transport and device design, but the generation and control of this peculiar form of bulk flow, particularly in higher dimensions, remain experimentally unexplored.

In this work, we experimentally demonstrate the tuning of directional transport in photonic quantum walks on a synthetic two-dimensional square lattice. The oriented bulk dynamics underlies the NHSE of the two-dimensional quantum walk—the unidirectional flow leads to the accumulation of eigenstates toward boundaries in the corresponding direction. By tuning the photon-loss parameters, we show how the direction of the flow (hence the direction of the NHSE) can be continuously adjusted. In particular, when the directional flow is tuned to the diagonal of the square lattice, the system exhibits the much-discussed corner skin effect[22]. By engineering the quantum-walk setup, we also introduce a synthetic flux to the lattice[41–45], which we observe to suppress the directional dynamics. Such suppression is the result of the competition between two localization mechanisms: magnetic confinement and the NHSE[46,47]. Specifically, the synthetic magnetic flux gives rise to local cyclotron orbits, which are incompatible with the persistent bulk flow underlying NHSE[46]. We quantitatively characterize the tunability of the light propagation through loss and flux, and further demonstrate their impact on the dynamics of topological edge modes along the boundary. Our experiment confirms the magnetic suppression of NHSE and further illustrates the flexible control over the NHSE-related bulk flow in higher dimensions.

[1]Beijing Computational Science Research Center, 100084 Beijing, China. [2]CAS Key Laboratory of Quantum Information, University of Science and Technology of China, 230026 Hefei, China. [3]CAS Center For Excellence in Quantum Information and Quantum Physics, 230026 Hefei, China. ✉e-mail: wyiz@ustc.edu.cn; gnep.eux@gmail.com

## Results

### Time-multiplexed two-dimensional quantum walk

In discrete-time quantum walks, the walker state $|\psi(t)\rangle$ evolves according to $|\psi(t)\rangle = U^t|\psi(0)\rangle$, where $t$ indicates the discrete time steps, and $U$ is thus identified as the Floquet operator that periodically drives the system. We consider such a quantum walk on a two-dimensional square lattice, with the Floquet operator

$$U = M_y S_y P C M_x S_x C. \qquad (1)$$

Here the shift operators are defined as $S_j = \sum_{\mathbf{r}} |0\rangle\langle 0| \otimes |\mathbf{r} - \mathbf{e}_j\rangle\langle\mathbf{r}| + |1\rangle\langle 1| \otimes |\mathbf{r} + \mathbf{e}_j\rangle\langle\mathbf{r}|$, with $\mathbf{r} = (x, y) \in \mathbb{Z}^2$ labeling the coordinates of the lattice sites, $j \in \{x, y\}$, and $\mathbf{e}_x = (1, 0)$ and $\mathbf{e}_y = (0, 1)$. The shift operators move the walker in the corresponding directions, depending on the walker's internal degrees of freedom on the basis of $\{|0\rangle, |1\rangle\}$ (dubbed the coin states). These coin states are subject to rotations under the coin operator $C = \frac{1}{\sqrt{2}}\begin{pmatrix} 1 & 1 \\ 1 & -1 \end{pmatrix} \otimes \mathbb{1}_{\mathbf{r}}$, where $\mathbb{1}_{\mathbf{r}} = \sum_{\mathbf{r}} |\mathbf{r}\rangle\langle\mathbf{r}|$. The gain–loss operators are given by (here $j \in \{x, y\}$)

$$M_j(\gamma_j) = \begin{pmatrix} e^{\gamma_j} & 0 \\ 0 & e^{-\gamma_j} \end{pmatrix} \otimes \mathbb{1}_{\mathbf{r}}, \qquad (2)$$

which makes the quantum walk non-unitary for finite $\gamma_x$ or $\gamma_y$.

A key ingredient to our scheme is the phase-shift operator, defined as

$$P = \sum_{\mathbf{r}} \begin{pmatrix} e^{i2\pi\alpha x} & 0 \\ 0 & e^{-i2\pi\alpha x} \end{pmatrix} \otimes |\mathbf{r}\rangle\langle\mathbf{r}|, \qquad (3)$$

which enforces a position-dependent geometric phase on the walker, so that the latter acquires a phase $2\pi\alpha$ when going around any single plaquette of the square lattice (see Fig. 1a and see the "Methods" section). Similar to that of the Hofstadter model[41], the accumulated phase shift of the walker on the lattice is equal to the Aharonov–Bohm phase of a charged particle in a uniform magnetic field, with a magnetic flux $\alpha$ threaded through each plaquette. We therefore regard $\alpha$ as the synthetic flux, which takes value in the range $[0, 1)$.

We experimentally implement the two-dimensional quantum walk above using photons. As illustrated in Fig. 1, the overall architecture is that of a fiber network[39,40,48–51], through which attenuated single-photon pulses are sent, with each full cycle around the network representing a discrete time step. The coin states $\{|0\rangle, |1\rangle\}$ are encoded in the photon polarizations $\{|H\rangle, |V\rangle\}$. The spatial degrees of freedom of the square lattice are encoded in the time domain, following a time-multiplexed scheme. This is achieved by building path-dependent time delays into the four different paths (labeled $x \pm 1$ and $y \pm 1$ in Fig. 1a) within the network (see the "Methods" section for details). The superpositions of multiple well-resolved pulses within the same discrete time step thus represent those of multiple spatial positions at the given time step (see Fig. 1b).

The shift and coin operators are implemented with beam splitters (BSs) and wave plates (WPs), and the phase operator with one of the electro-optical modulators (EOM1 in Fig. 1). We further implement polarization-dependent loss operators $M'_j = e^{-\gamma_j} M_j$ in each path, using a combination of the WPs and the EOMs. The time-evolved state driven by $U$ is then related to that in the experiment by adding a factor $e^{(\gamma_x + \gamma_y)t}$ to the latter.

For all experiments, avalanche photo-diodes (APDs) with temporal and polarization resolutions are employed to record the probability distribution of the walker states. This enables us to construct

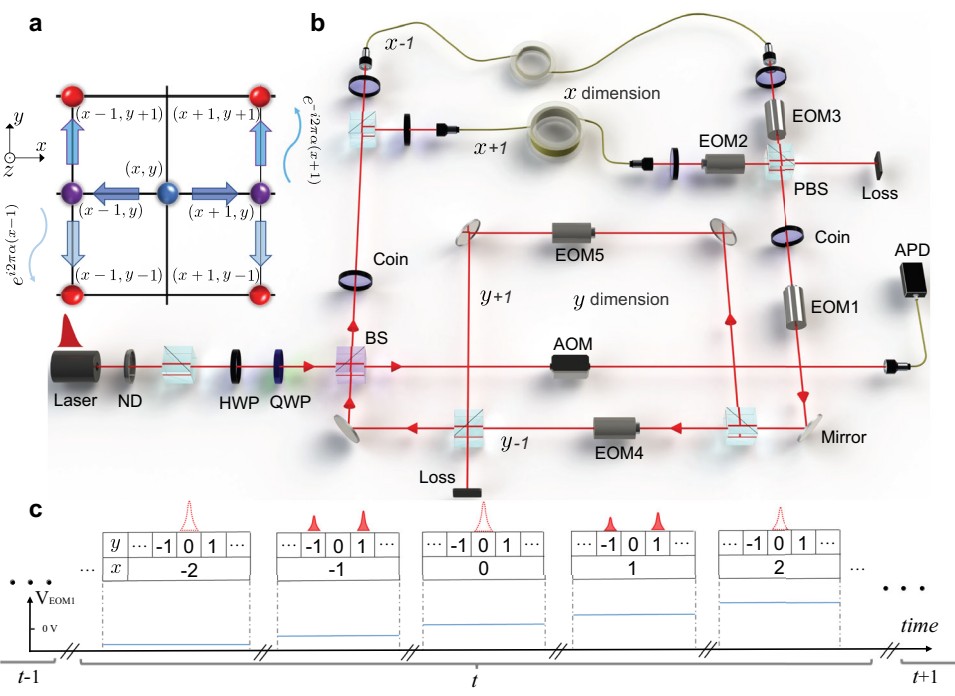

**Fig. 1 | Two-dimensional non-Hermitian quantum walk with a synthetic gauge field. a** Schematics for possible movements of a walker at spatial position $(x, y)$ during each time step. **b** A time-multiplexed implementation of the two-dimensional photonic quantum walk. The photons are initialized at position $(0, 0)$ in the superposition of the polarizations $(|H\rangle + i|V\rangle)/\sqrt{2}$. Once coupled into the setup through a low-reflectivity beam splitter (BS, reflectivity 3%), their polarization state is manipulated by a half-wave plate (HWP). The photonic wave packets are split by a polarizing beam splitter (PBS) and routed through a pair of single-mode

fibers (SMF) of length 287.03 and 270 m, respectively, implementing a temporal step in the $x$ direction. A temporal step in the $y$ direction is implemented by another two-PBS loop based on the same principle but in the free space instead of fibers. At each step, photons are partially coupled out to a polarization resolving detection of the arrival time via avalanche photodiodes (APDs). ND neutral density filter, AOM optical switch acousto-optic modulator, EOM electro-optic modulator. **c** Illustration of the operation sequence of the time-multiplexed quantum walk. Here $V_{EOM}$ is the control voltage applied to the EOMs.

the site-resolved population of the synthetic lattice, with

$$P_{exp}(x,y,t) = \frac{N(x,y,t)}{\sum_{x,y} N(x,y,t)}, \qquad (4)$$

where $N(x,y,t)$ is the total photon number on site $(x,y)$ at time $t$.

Before discussing our experimental observations, two remarks are in order. First, by using attenuated laser pulses, our experiment is performed in the classical regime, as we simulate the dynamics and interference of single photons using the coherence of laser pulses. Nevertheless, what we implement here can still be regarded as quantum walk, not only because the term is widely adopted for similar setups in the literature[48–51]. More importantly, it highlights the difference between our experiment and that of classical random walk. While quantum walks are deterministic in their time evolution (driven repeatedly by the Floquet operator), a random walk is intrinsically stochastic. The two also differ drastically in the spreading of the walker distribution[52,53].

Second, the overall time-multiplexed setup and the synthetic-flux engineering are similar to those reported in ref. 43 where purely dissipative nearest-neighbor couplings are implemented. However, while ref. 43 exactly built a tight-binding model, for our discrete-time quantum-walk setup, a transparent tight-binding perspective can only be achieved in the high-frequency limit (see the "Methods" section).

## NHSE and tunable photon transport

In the absence of flux, quantum walks driven by $U$ already show directional transport under finite photon losses. In Fig. 2a, we show the measured populations of the synthetic lattice sites after $t = 16$ time steps. Starting from a local initial state at $\mathbf{r} = (0, 0)$, the propagation in the synthetic spatial dimensions is symmetric along the four lattice directions (Fig. 2a). However, under finite photon-loss parameters, the final-time photon distribution becomes asymmetric with a preferred direction. For instance, when $\gamma_x = \gamma_y \neq 0$, as shown in Fig. 2b, the flow is diagonal to the square lattice. By tuning the ratio of $\gamma_y/\gamma_x$, we can continuously adjust the direction of the asymmetric pattern. This is explicitly shown in Fig. 2c, where we define the directional displacement

$$\mathbf{d}(t) = \sum_{x,y} \mathbf{r} P_{exp}(x,y,t). \qquad (5)$$

As $\gamma_y/\gamma_x$ varies, the direction of the displacement at the final time step can be continuously tuned (see the left panel of Fig. 2c). In our experiment, we adjust $\gamma_y/\gamma_x$ in the range of $[-1, 1]$ for 16-time-step quantum walks. Correspondingly, the measured polar angle of $\mathbf{d}$ changes from $3\pi/4$ to $-3\pi/4$ (the right panel of Fig. 2c).

The observed loss-induced directional flow is closely related to the NHSE in two dimensions. While it is straightforward to show that the orientation of the directional transport also indicates the direction of the eigenstate accumulation under the open boundary condition

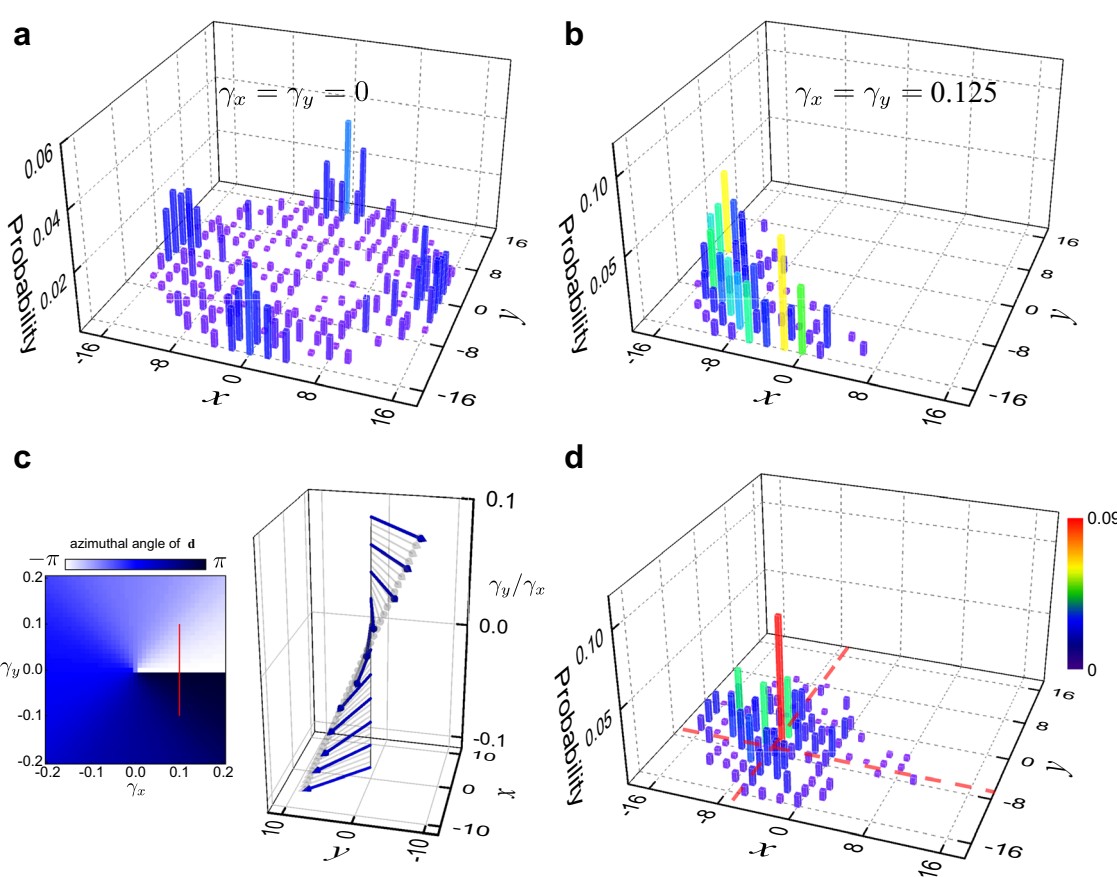

**Fig. 2 | Tunable directional flow and NHSE.** The walker with the polarization $(|H\rangle + i|V\rangle)/\sqrt{2}$ starts from the lattice site $(0, 0)$ with $\alpha = 0$. Probability distributions are measured after 16 time steps. **a** Probability distribution for a Hermitian two-dimensional quantum walk with $\gamma_x = \gamma_y = 0$. **b** Probability distribution following a non-Hermitian quantum walk with $\gamma_x = \gamma_y = 0.125$. **c** Directional displacements after the final time step ($t = 16$) for quantum walks with varying $\gamma_x$ and $\gamma_y$. (Left) Color contour of the azimuthal angle of the displacement $\mathbf{d}$ on the $x$–$y$ plane. (Right) Measured (blue arrows) and simulated (gray arrows) the displacement along the red vertical line of the color contour (left panel). **d** Probability distribution following a non-Hermitian quantum walk in the presence of domain walls (marked by red dashed lines), with $\gamma_x = 0.125$ for $x \geq -6$, $\gamma_x = -0.125$ for $x < -6$, $\gamma_y = 0.125$ for $y \geq -6$, and $\gamma_y = -0.125$ for $y < -6$.

(see Supplementary Note 2), from an experimental perspective, we observe the dynamic localization of the walker toward the boundary when a domain-wall boundary condition is imposed (see Fig. 2d). Combined with the theoretical spectral analysis that there are no topological edge states present under the parameters of Fig. 2d, it is clear that the localization is due to the NHSE.

### Magnetic suppression of the directional transport

In Fig. 3a–d, we show the final population distribution with the synthetic flux switched on, following 16-time-step quantum walks. Compared to Fig. 2, the directional flow appears to be increasingly suppressed under larger $\alpha$, regardless of its direction. In Fig. 3e, f, we show the absolute values of the directional displacement **d** as functions of $\alpha$, for various loss parameters. The suppression is the largest when $\alpha$ is tuned in between 0 and 0.5. Such a suppression reflects the competition between the magnetic confinement and the persistent bulk flow and can be used for the manipulation of the photon transport.

Note that we only plot the measurements for $\alpha \in [0, 0.5]$, since the on-site-occupation dynamics (and hence the directional displacement) are the same under $\alpha$ and $\alpha + 0.5$ (see Supplementary Note 3). Furthermore, the underlying competition between the magnetic confinement and NHSE can be clarified by numerically analyzing the response of eigenspectra and eigenstate distribution to the magnetic flux (see Supplementary Note 2).

### Impact on topological edge states

In the absence of loss, the Floquet operator $U$ describes an anomalous Floquet Chern insulator, characterized by the Floquet topological invariant[45,54,55], which can be calculated for each quasienergy gap and is fully responsible for the topological edge states. Here we experimentally investigate how the NHSE under loss and magnetic confinement impacts the topological edge states. For this purpose, we engineer a

domain-wall configuration by choosing different values of $\alpha$ on either side of $x = 0$.

As shown in Fig. 4a, for lossless quantum walks, a pair of topological edge modes emerge, moving in opposite directions along the boundary. This is consistent with the prediction of the Floquet topological invariant (see the "Methods" section and Supplementary Note 2). When only the loss parameter $\gamma_x$ is turned on, the NHSE induces a horizontal directional flow toward the region with $x < 0$. From the measured population following a 16-time-step quantum walk (see Fig. 4b), both the bulk flow and the topological edge states are clearly visible. Since the directional flow is perpendicular to the boundary, it has no direct impact on the motion of the topological edge states. This is no longer the case when both $\gamma_x$ and $\gamma_y$ become finite, as in Fig. 4c. Here, besides a diagonal bulk flow indicating the corner skin effect, the topological edge modes moving in the negative (positive) $y$ direction are enhanced (suppressed) by the NHSE.

In Fig. 4d–f, we show the final probability distribution under a larger synthetic flux $\alpha$. Compared to Fig. 4a–c, the bulk propagation is significantly suppressed, whereas the topological edge modes are largely unaffected by flux. This suggests that magnetic confinement is helpful for the dynamics detection of topological edge modes in systems with the NHSE.

## Discussion

We have experimentally demonstrated how the interplay of synthetic flux and dissipation enables full control over the directional transport underlying the NHSE. Since the quantum walk simulates an anomalous Floquet Chern insulator, we further illustrate how the motion of topological edge modes on the boundary is affected by the tuning parameters. While the high tunability can be exploited for topological device design, our implementation of a dissipative anomalous Floquet Chern insulator further raises theoretical questions as to how the NHSE affects the bulk-boundary correspondence herein. Our experiment also paves

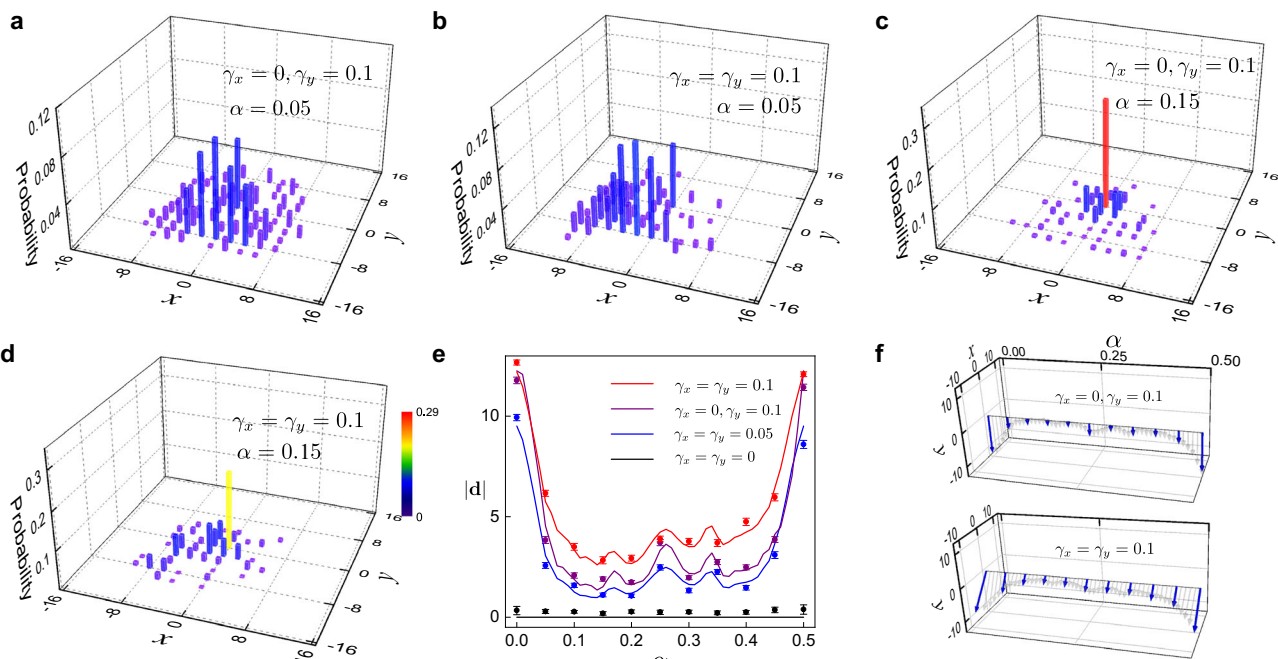

**Fig. 3 | Magnetic suppression of the directional transport.** The walker is initialized in the state $\frac{1}{\sqrt{2}}(|H\rangle + i|V\rangle) \otimes |x=0\rangle|y=0\rangle$. Measured probability distributions of 16-time-step quantum walks with the tuning parameter $\alpha = 0.05$, and the loss parameter $\gamma_x = 0$, $\gamma_y = 0.1$ in **a**, with $\alpha = 0.15$, $\gamma_x = 0$, and $\gamma_y = 0.1$ in **b**, with $\alpha = 0.05$ and $\gamma_x = \gamma_y = 0.1$ in **c**, and with $\alpha = 0.15$ and $\gamma_x = \gamma_y = 0.1$ in (**d**). **e** Norm of the directional displacement **d** for 16-time-step quantum walks under different flux $\alpha$. Symbols

represent the experimental data, and curves are the corresponding numerical simulations. Error bars are due to the statistical uncertainty in photon-number-counting. **f** Measured (blue) and simulated (gray). **d** Blue arrows represent the experimental results, and gray ones indicate the corresponding numerical simulations.

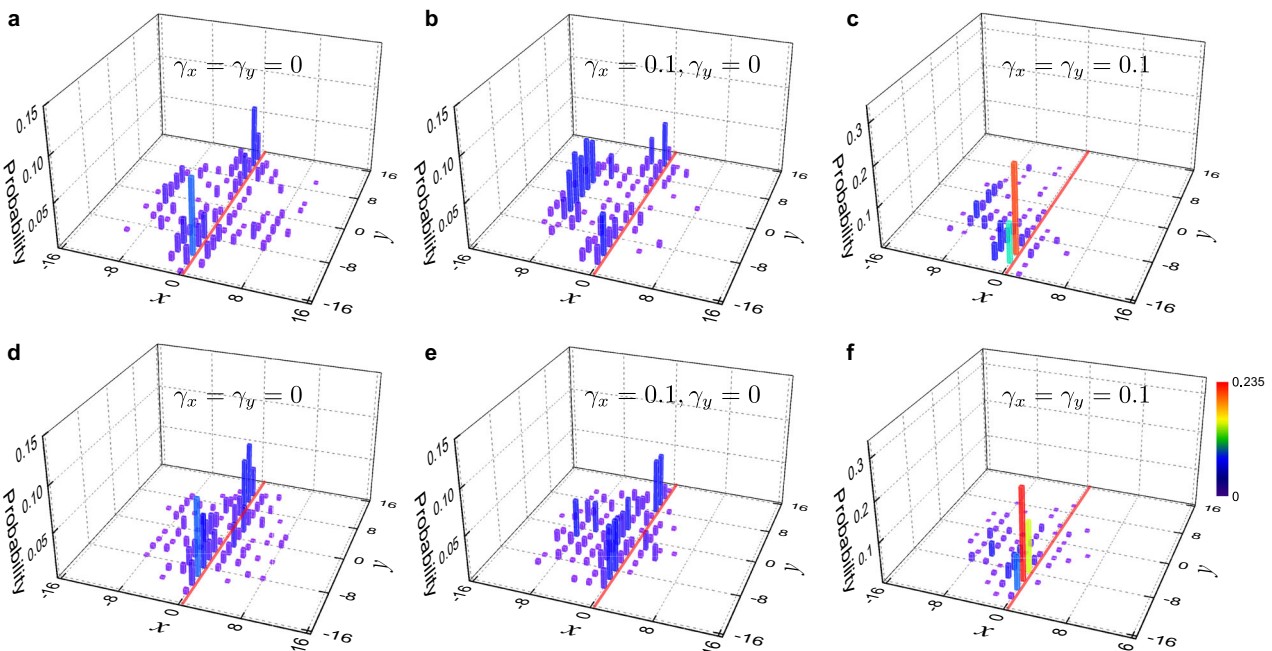

**Fig. 4 | Dynamic detection of topological edge modes in 16-time-step two-dimensional magnetic quantum walks.** The walker is initialized in the state $\frac{1}{\sqrt{2}}(|H\rangle + i|V\rangle) \otimes |x=0\rangle|y=0\rangle$ under a domain-wall geometry. Specifically, in **a–c**, we set $\alpha = 0.05$ for $x \leq 0$, and $\alpha = -0.05$ for $x > 0$. In **d–f**, we set $\alpha = 1/3$ for $x \leq 0$, and $\alpha = -1/3$ for $x > 0$. The left, middle, and right columns show the measured probability distribution with the parameters $\gamma_x = \gamma_y = 0$, $\gamma_x = 0.1$, $\gamma_y = 0$, and $\gamma_x = \gamma_y = 0.1$, respectively.

the way for engineering more exotic forms of the non-Hermitian skin effect in higher dimensions[22] using quantum-walk dynamics.

## Methods
### Experimental setup
We adopt a time-multiplexed scheme for the experimental realization of photonic quantum walks[39,40,48–51]. As illustrated in Fig. 1, the photon source is provided by a pulsed laser with a central wavelength of 808 nm, a pulse width of 88 ps, and a repetition rate of 15.625 kHz. The pulses are attenuated by a neutral density filter, such that an average photon number per pulse is less than $2.4 \times 10^{-4}$, which ensures a negligible probability of multi-photon events. The photons are coupled in and out of a time-multiplexed setup through a BS with a reflectivity of 3%, corresponding to a low coupling rate of photons into the network. Such a low-reflectivity BS also enables the out-coupling of photons for measurement. A HWP with the setting angle $\pi/8$ is used to implement the coin operator $C$.

Four different paths in a fiber network correspond to the four different directions a walker can take in one step on a two-dimensional lattice. Two-PBS loops are used to realize polarization-dependent optical delays. The shift operator $S_x$ is implemented by separating photons corresponding to their two polarization components and routing them through the fiber loops, respectively. Polarization-dependent time delay is then introduced. Since the lengths of the two fiber loops are 287.03 and 270.00 m, respectively, the time difference of photons traveling through two fiber loops is 80 ns. The shift operator $S_y$ is implemented by another two-PBS loop based on the same principle, where the vertical component of photons is delayed relative to the horizontal component by a 1.61 m free space path difference. The corresponding time difference in the $y$ direction is then 4.83 ns.

The position-dependent phase operator $P$ is implemented using the first electro-optical modulator (EOM1). The rise/fall times of EOM (4 ns) are much shorter than the time difference between adjacent positions (80 and 4.83 ns for $x$ and $y$ directions, respectively), which enables us to control the parameter $\phi$ precisely.

To realize a polarization-dependent loss operation $M'_x(\gamma_x) = e^{-\gamma_x}M_x(\gamma_x)$, two HWPs and an EOM are introduced into each fiber loop. Here HWPs are used to keep the polarizations of photons unchanged before and after they pass through the fiber loops. For $\gamma_x > 0$, for the short loop, the voltage of EOM3 is tuned to 0. Thus, after passing through the first PBS, horizontally polarized photons are all transmitted by the second PBS and are subject to further time evolution. Whereas for the long loop, by controlling the voltage of EOM2 to satisfy $\cos\theta/2 = e^{-2\gamma_x}$, we flip part of the photons $(1 - e^{-4\gamma_x})$ with vertical polarization into horizontal ones. They are subsequently transmitted by the second PBS and leak out of the setup. Otherwise, for $\gamma_x < 0$, horizontally polarized photons are all transmitted by the second PBS and are subject to further time evolution in the long loop. By contrast, for the short loop, part of the photons $(1 - e^{4\gamma_x})$ with vertical polarization is flipped by EOM3, transmitted by the second PBS and subsequently leaked out of the setup. We use the same method to realize $M'_y(\gamma_y)$.

To measure the probability distribution, we switch on the AOM for 4.9 µs following a given time step. The time-resolved pulses within this window are then recorded and translated to the corresponding spatial position. The measurement time for a specific time step is one and a half hours, limited by the stability of our setup. The environmental noise and those that originate from the dark counts of the detectors are uniformly distributed in time and are easily eliminated during data analysis.

We compare the ideal theoretical distribution with the measured distribution via the similarity,

$$S(t) = \sum_{x,y} \sqrt{P_{\text{th}}(x,y,t)P_{\text{exp}}(x,y,t)}, \qquad (6)$$

which quantifies the equality of two probability distributions. Here $S = 0$ stands for completely orthogonal distributions, and $S = 1$ for identical distributions. We observe $S \geq 0.914$ in Fig. 2, $S \geq 0.922$ in Fig. 3, and $S \geq 0.930$ in Fig. 4, respectively. Here the theoretical value

$P_{\text{th}}(x, y, t)$ is given by

$$P_{\text{th}}(x,y,t) = \sum_{m=H,V} \frac{|(\langle \mathbf{r}| \otimes \langle m|)e^{-(\gamma_x + \gamma_y)t}|\psi(t)\rangle|^2}{\sum_{\mathbf{r},m}|(\langle \mathbf{r}| \otimes \langle m|)e^{-(\gamma_x + \gamma_y)t}|\psi(t)\rangle|^2}, \tag{7}$$

where $|\psi(t)\rangle$ is the time-evolved walker state under $U$.

In our experiment, photon loss is caused by the loss of photons through an optical element. Our round-trip single-loop efficiency is about 0.66 even for a unitary quantum walk. This is calculated by multiplying the transmission rates of each optical component used in the round trip, including the transmission rates of the BS (-0.97), the collection efficiency from free space to fiber (-0.78), the EOM (-0.96), and all other optical components (-0.91). We therefore estimate the single-loop efficiency as $0.78 \times 0.97 \times 0.96 \times 0.91 \simeq 0.66$.

## Floquet topological invariant

The walker state evolves according to

$$|\psi(t)\rangle = U^t |\psi(0)\rangle = e^{-iH_{\text{eff}}t}|\psi(0)\rangle, \tag{8}$$

where $H_{\text{eff}} = i \ln U$ is defined as the effective Hamiltonian. While the quantum walk is identified as the periodically driven Floquet dynamics, the eigenenergies of $H_{\text{eff}}$ constitute the quasienergy spectrum of the Floquet system. We fix the branch cut of the logarithm such that the quasienergy spectrum lies within the range $[-\pi, \pi)$.

To calculate the Floquet topological invariant, we follow refs. [45], [54], [55], and define

$$U' = e^{i\tilde{E}}M_y S_y P'CM_x S_x C, \tag{9}$$

where $e^{i\tilde{E}}$ shifts the quasienergy spectrum by $-\tilde{E}$. The modified phase-shift operator is

$$P'(\beta,\alpha) = \sum_{\mathbf{r}} \exp\left[i\sigma_z(\beta\lfloor x/q\rfloor + 2\pi\alpha x)\right] \otimes |\mathbf{r}\rangle\langle \mathbf{r}|, \tag{10}$$

where $\alpha = p/q$, $\beta = 2\pi/s$, and $\lfloor x/q \rfloor$ is the greatest integer less than or equal to $x/q$. Here $p$ and $q$ are coprime integers, and $s$ is a sufficiently large integer (in our case, for $\alpha = 1/3$, $s = 15$ is sufficient).

We denote the eigenvalues of $U'$ as $e^{-iE_j}$, and the topological invariant for the quasienergy gap (corresponding to $U$) comprising $\tilde{E}$ can be calculated through

$$R = \frac{1}{2\pi}\left(\sum_{j=1}^{2sq} E_j(1/s,\alpha,\tilde{E}) - \sum_{j=1}^{2sq} E_j(0,\alpha,\tilde{E})\right). \tag{11}$$

Under an open boundary condition, the value of $R$ indicates the number of anomalous Floquet edge states emerging within the quasienergy gap.

In Fig. 2d, all quasienergy gaps are closed, hence there are no topological edge states along the boundaries, and the gap topological invariants are ill-defined. In Fig. 4, for the $x \leq 0$ ($x > 0$) region of the domain-wall configurations, we have $p = 1, q = 20$ ($p = -1, q = 20$) in Fig. 4a–c, and $p = 1, q = 3$ ($p = -1, q = 3$) in Fig. 4d, e, f, respectively. While there are now a host of quasienergy gaps, for any given gap, the topological invariants $R$ of the two regions are always finite and differ by a sign (see Supplementary Note 4). As a consequence, Floquet topological edge modes emerge along the domain-wall boundary.

We find that the Floquet topological invariant $R$ is capable of predicting the anomalous topological edge states under all our experimental parameters, despite the presence of the NHSE. Whether the NHSE can have a significant impact on $R$ beyond our experimental parameters (particularly when the photon loss is further increased) is an interesting theoretical question that we leave to future studies.

## A tight-binding perspective

In general, for a discrete-time quantum walk driven by $U$, the effective Hamiltonian can be defined as $U = e^{-iH_{\text{eff}}}$, such that the quantum walk constitutes a stroboscopic simulation of the time evolution governed by $H_{\text{eff}}$. While $H_{\text{eff}}$ can be formally derived, it is typically complicated and does not yield much insight.

Nevertheless, we can work in the high-frequency limit, assuming each time step to be short enough that the time evolution can be Trotterized and kept to the lowest order. More explicitly, we define the effective Hamiltonians for the following combinations of components in $U$, with

$$C = e^{-iH_1}, \quad M_x S_x = e^{-iH_2}, \quad M_y S_y P = e^{-iH_3}. \tag{12}$$

The time evolution within each time step is then approximately driven by the effective Hamiltonian

$$H'_{\text{eff}} = \frac{1}{4}(2H_1 + H_2 + H_3), \tag{13}$$

where

$$H_1 = \frac{\pi}{2\sqrt{2}}\begin{pmatrix} 1 - \sqrt{2} & 1 \\ 1 & -(1+\sqrt{2}) \end{pmatrix} \otimes \mathbb{1}_{\mathbf{r}}, \tag{14}$$

$$H_2 = \sum_{x,y}\left[e^{\gamma_x}\begin{pmatrix} \frac{i}{2} & 0 \\ 0 & -\frac{i}{2} \end{pmatrix} \otimes |x-1,y\rangle\langle x,y| \right.$$
$$\left. -e^{-\gamma_x}\begin{pmatrix} \frac{i}{2} & 0 \\ 0 & -\frac{i}{2} \end{pmatrix} \otimes |x,y\rangle\langle x-1,y|\right], \tag{15}$$

$$H_3 = \sum_{x,y}\left[e^{i2\pi\alpha x}e^{\gamma_y}\begin{pmatrix} \frac{i}{2} & 0 \\ 0 & -\frac{i}{2} \end{pmatrix} \otimes |x,y-1\rangle\langle x,y| \right.$$
$$\left. -e^{-i2\pi\alpha x}e^{-\gamma_y}\begin{pmatrix} \frac{i}{2} & 0 \\ 0 & -\frac{i}{2} \end{pmatrix} \otimes |x,y\rangle\langle x,y-1|\right]. \tag{16}$$

Here $H_2$ and $H_3$, respectively, describe asymmetric hopping along the $x$ and $y$ directions, which can be considered as the microscopic origin of the NHSE and the directional flow. In addition, a Peierls phase emerges in the hopping terms in $H_3$, which enforces the synthetic magnetic flux illustrated in Fig. 1a. Thus, while $H'_{\text{eff}} \neq H_{\text{eff}}$ and does not fully account for the discrete-time quantum-walk dynamics, it offers an approximate, but more transparent understanding of the quantum-walk dynamics from the perspective of a tight-binding model.

## Data availability
The data that support the findings of this study are available from the corresponding authors.

## Code availability
The codes that support the findings of this study are available from the corresponding authors.

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

## Acknowledgements

We thank Chen Fang for the helpful discussions. This work has been supported by the National Natural Science Foundation of China (Grant Nos. 92265209, 12025401, 11974331, and 12088101).

## Author contributions

Q.L. performed the experiments. W.Y. developed the theoretical aspects and performed the theoretical analysis, and wrote part of the paper. P.X. supervised the project, designed the experiments, analyzed the results, and wrote part of the paper.

## Competing interests

The authors declare no competing interests.
