## [Peer Review File · Nature Communications]

Manipulating directional flow in a two-dimensional photonic quantum walk under a synthetic magnetic fieldREVIEWER COMMENTS

Reviewer #1 (Remarks to the Author):

The MS presents a clear experiment in the classical domain. The results are OK but without a significant explanation of what are the physical mechanisms for such observation.

Here are detail comments:

1-In the abstract, it is written that: "The non-reciprocal dynamics hereof is a manifestation of the non-Hermitian skin effect, with its direction continuously adjustable through the photon-loss parameters."

I am not sure if this statement is true. Skin effect is in fact a consequence of non-reciprocity.

2- Then it is written that "By contrast, the synthetic flux originates from an engineered geometric phase, which competes with the non-Hermitian skin effect through magnetic confinement. "

I could not how the synthetic flux competes with NHSE. In what sense?

3- While the introduction starts with quantum systems none of the refs. 2-9 are quantum systems.

4- Complicated statement with no fine meaning: "The much discussed corner skin effect in two dimensions is but a special case here, corresponding to ... "

5- I am against using the term "non-reciprocal" specifically when there is no "magnetic flux" . There is nothing there to be non-reciprocal. The system shows an assymmetric pattern which is correctly explained in the paper. But calling it non-reciprocal is not correct. So this statement needs to be removed "Such a directional flow is the signature of the non-reciprocal transport. "

6- I could not find a good explanation for the section "magnetic suppression of NHSE". The authors just show what they observe without any supporting discussion. For example, they could show the mode competition or mode distribution. Also why α in (0,0.5) shows

the largest suppression? why not (0,0.75)? There is no explanation of how this range becomes important.

7- I recommend not using "quantum" as there is nothing in the paper about quantum. The experiment is a random walker in a discrete lattice. Thus, it is misleading to use the term quantum.

In summary, the abstract needs a lot of work as it is not clearly written. The results are presented nicely but there is a significant lack of analytical and quantitative explanations for the observations. There are also technical terms that need to be used/removed as they provide the wrong message.

Reviewer #2 (Remarks to the Author):

This work reports the realization of non-Hermitian synthetic lattices supporting NHSE in quantum-walk experiments and, by adding effective (synthetic) magnetic flux, the influence of the flux on NHSE is experimentally explored. The subject involving quantum walk, effective magnetic flux, and also NHSE certainly attracts broad interest in both optic and quantum communities. The experimental results are clear to support the conclusions, especially the magnetic suppression of NHSE is of great importance for future flexible control of light (photons) in higher dimensions. I think this study should be published in high impact journal such as Nat. Commun. Here I list some comments for the authors to consider:

1. It should be noticed (as far as I understand), Ref. [43] may have different mechanism as that in the current manuscript to build the synthetic lattice. The key difference is that delay lines in Ref. [43] guide only a small portion of the energy in each pulse so the experimental platform exactly builds a tight-binding lattice. However, in the current work, the platform supports two-dimensional photonic quantum walk. Firstly, I think this difference should be acknowledged. Secondly, although I can understand how such 2D quantum walk works, it is not very clear to me how it can be understood in a tight-binding model as shown in Fig. 1a. Further clarification is appreciated.

2. In Fig. 3e, I see symmetric patterns with alpha from 0 to 0.5. Is there any explanation, as the flux is $2\pi\alpha$ and one may expect to see symmetric patterns with alpha from 0 to 1.

3. The authors envision that their experiment can help study NHSE in higher dimensions. However, to the best of my knowledge, the current experimental platform is limited in the total number of time bins. If three dimensional lattice is to be studied, then the number of time bins is largely increased. Can the authors provide some perspective on this matter? Or in other words, will the researchers go to three dimensions in the near future?

4. Although I am usually fine with the phrase 'quantum walk', I do not see much 'quantum' here. The experiment is conducted with classical light (if not, please correct me). Such phenomena can also be named as 'random walk'. Probably it will be good to discuss why this platform is 'quantum' and can be used for quantum device design.

Reply to Report of Referee A

We thank the Referee for her/his valuable time, stimulating comments, and helpful suggestions. In the following, let us address these questions one-by-one.

The MS presents a clear experiment in the classical domain. The results are OK but without a significant explanation of what are the physical mechanisms for such observation. Here are detail comments:

We thank the Referee for finding our experiment clear. In this revision, we made an effort to provide further analytical derivation and numerical simulation, which help to clarify the underlying mechanism of our observations. We hope the Referee will be satisfied with these additions.

1- In the abstract, it is written that: “The non-reciprocal dynamics hereof is a manifestation of the non-Hermitian skin effect, with its direction continuously adjustable through the photon-loss parameters.”

I am not sure if this statement is true. Skin effect is in fact a consequence of non-reciprocity.

The non-Hermitian skin effect is typically defined as the accumulation of a macroscopic number of eigenstates toward the boundary, and is often attributed to the spectral topology of the system under the periodic boundary condition (see Refs. [22,31,32]). Mathematically, the origin of these phenomena is the sensitivity of the eigenvectors and eigenvalues of non-Hermitian matrices to boundary perturbations (see Ref. [5]). In this sense, the deformation of eigenstates under the open boundary condition, the spectral topology under the periodic boundary condition, and the directional bulk flow we observe here, can all be traced back to the same origin, and are closely related to one another. In our previous manuscript, we referred to this broad class of phenomena as the non-Hermitian skin effect, which can be confusing indeed. In this resubmission, we revised the sentence to “The directional flow hereof underlies the emergence of the non-Hermitian skin effect...”.

We note that non-reciprocity also exists in non-Hermitian systems without the non-Hermitian skin effect [c.f. Phys. Rev. B **100**, 054301 (2019); Phys. Rev. B **105**, 245407 (2022); Phys. Rev. A **107**, 022222 (2023); Phys. Rev. X **5**, 021025(2015)], or even in

Hermitian systems [c.f. *Opt. Lett.* **47**, 5437-5440 (2022)]. We therefore prefer not to make a general claim that the non-Hermitian skin effect is a consequence of non-reciprocity (or of the directional flow), for fear of misunderstanding.

2- Then it is written that “By contrast, the synthetic flux originates from an engineered geometric phase, which competes with the non-Hermitian skin effect through magnetic confinement.” I could not how the synthetic flux competes with NHSE. In what sense?

We thank the Referee for the stimulating question, which gives us the opportunity to clarify the physical mechanism behind our observation.

The competition between the non-Hermitian skin effect and the magnetic confinement refers to the boundary localization under the non-Hermitian skin effect, versus the magnetic confinement that leads to localized eigenstates in the bulk. In terms of dynamics, the competition gives rise to the suppressed directional flow as observed in our experiment.

As illustrated in Fig. 1a, the walker acquires a position-dependent phase when moving along the y direction, thanks to the operator sequence $S_y P$ in the Floquet operator U . This process contributes a position-dependent phase to the corresponding hopping terms in the high-frequency effective Hamiltonian (see derivation and discussions in the revised Methods). As a consequence, when the walker goes around the plaquette in a unit cell, it accumulates a non-vanishing phase $2\pi\alpha$. The position-dependent phase on the hopping terms is formally the same as the Peierls phase, which appears in the hopping terms of charged particles under a magnetic field [c.f. *Z. Phys.* **80**, 763 (1933); *Phys. Rev. A* **92**, 042324 (2015)]. As such, our quantum-walk dynamics is akin to the dynamics of a charged particle moving in a two-dimensional lattice, perpendicular to a uniform magnetic field. The synthetic flux α then corresponds to the magnetic flux threaded through each unit cell.

Now, for a charged particle moving in a plane perpendicular to a uniform magnetic field, it experiences a force perpendicular to both its direction of motion and the magnetic field, leading to cyclotron motion with localized orbits. The quantization of these cyclotron orbits gives the well-known flat Landau bands with localized eigenstates. Since such a localization occurs in the bulk, it is incompatible, and competes with the non-Hermitian skin effects which deform the eigenstates toward the boundary. The dynamic manifestation of such a competition is the flux-induced suppression of the direction flow, as we reported in the

experiment.

To gain further insight into the competition, we plot the quasienergy spectra under different magnetic fluxes in Fig. R1. For this purpose, we define the effective Hamiltonian through $U = e^{-iH_{\text{eff}}}$, and calculate the eigenspectra of H_{eff} . When $\alpha = 0$, the eigenspectra occupy a finite area in the complex plane under the periodic boundary condition, whereas the eigenspectra collapse (occupying zero area) under the open boundary condition. The spectral topology under the periodic boundary condition is a key indicator of the non-Hermitian skin effect in two dimensions (see Ref. [22]). Dynamically, the area occupied by the eigenspectra corresponds to the short-time acceleration of the directional flow [c.f. Phys. Rev. B **105**, 245143 (2022)]. However, under a finite α , such an area shrinks significantly (see Fig. R1b), indicating the flux-induced suppression of the directional flow, and the non-Hermitian skin effect as well.

The observation is further corroborated by the spatial distribution of eigenstates. We define the averaged spatial distribution of eigenstates under the open boundary condition

$$|\overline{\psi(x)}| = \sum_i^{Ne} \sum_y \frac{|\psi_i(x, y)|^2}{Ne}, \quad (\text{R1})$$

$$|\overline{\psi(y)}| = \sum_i^{Ne} \sum_x \frac{|\psi_i(x, y)|^2}{Ne}, \quad (\text{R2})$$

where $|\psi_i(x, y)|^2$ is the occupation of the i th eigenstate at site (x, y) , and Ne is the total number of eigenstates. As shown Figs. R1c and d, while the non-Hermitian skin effect is apparent in either spatial direction, it is suppressed by the presence of a magnetic flux. Our observed competition between the non-Hermitian skin effect and the magnetic confinement is also consistent with previous theoretical studies [c.f. Phys. Rev. Lett. **127**, 256402 (2021); Phys. Rev. B **106**, L081402 (2022)].

In the revised manuscript, we added more discussions on the origin of such a competition both in the main text and in the Supplemental Material.

3- While the introduction starts with quantum systems none of the refs. 2-9 are quantum systems.

We changed the opening sentence to “Open systems...”, which is more fitting for the general discussion of non-Hermitian physics. We thank the Referee for the comment.

FIG. R1. **a, b** Numerically calculated quasienergy spectra of H_{eff} on the complex plane under the periodic boundary condition with $\gamma_x = \gamma_y = 0.1$ under different α . **c, d** The corresponding averaged spatial distribution of eigenstate in x and y directions under the open boundary condition. For numerical calculations, we take a lattice with the size 33×33 .

4- Complicated statement with no fine meaning: “The much discussed corner skin effect in two dimensions is but a special case here, corresponding to ...”

For clarity, we changed the sentence to “In particular, when the directional flow is tuned to the diagonal of the square lattice, the system exhibits the much discussed corner skin effect [22].” Reference [22] is also provided at the end of the sentence to provide a proper context.

5- I am against using the term “non-reciprocal” specifically when there is no “magnetic flux”. There is nothing there to be non-reciprocal. The system shows an assymmetric pattern which is correctly explained in the paper. But calling it non-reciprocal is not correct. So this statement needs to be removed “Such a directional flow is the signature of the non-reciprocal transport.”

We removed the terms “non-reciprocal” and “non-reciprocity” throughout the text, and revised the title and abstract accordingly.

6- I could not find a good explanation for the section “magnetic suppression of NHSE”. The authors just show what they observe without any supporting discussion. For example, they could show the mode competition or mode distribution. Also why α in $(0, 0.5)$ shows the largest suppression? why not $(0, 0.75)$? There is no explanation of how this range becomes important.

In the revised manuscript, we provided supporting analysis on the magnetic suppression of the NHSE. As detailed in our response to the Referee’s second comment above, these include analytic derivation of the high-frequency Hamiltonian with the Peierls phase in the revised Methods; discussion and clarification of the physical mechanism behind the competition between the non-Hermitian skin effect and magnetic confinement in the revised Supplemental Material; and numerical calculations of the eigenspectra and eigenstates distribution in the revised Supplemental Material.

On the other hand, we chose the parameter range $\alpha \in [0, 0.5)$ because the dynamics of on-site occupation under $U(\alpha)$ and $U(\alpha + 0.5)$ are the same. To explicitly demonstrate this, we extend the parameter range of Fig. 3(e) in the main text, and show the numerically evaluated $|\mathbf{d}|$ in Fig. R2, with the observation that $|\mathbf{d}(\alpha)| = |\mathbf{d}(\alpha + 0.5)|$.

To understand this observation, we fit the Floquet operator U in a different time frame, i.e., subject it to a unitary transformation $\bar{U} = S_y^\dagger U S_y$, where S_y is the shift operator defined in the manuscript. Denoting $U_0 = M_y C M_x S_x C$, we have

$$\begin{aligned}\bar{U} &= P U_0 \\ &= P_{\text{odd}} U_0 + P_{\text{even}} U_0,\end{aligned}\tag{R3}$$

where

$$P_{\text{odd (even)}} = \begin{cases} \sum_{x,y} \begin{pmatrix} e^{i2\pi\alpha x} & 0 \\ 0 & e^{-i2\pi\alpha x} \end{pmatrix} \otimes |x, y\rangle \langle x, y|, & x \text{ is odd (even)}; \\ 0, & \text{otherwise.} \end{cases}\tag{R4}$$

Consider $\alpha' = \alpha + 0.5$, we have

$$\bar{U}(\alpha) = P_{\text{odd}}(\alpha) U_0 + P_{\text{even}}(\alpha) U_0,\tag{R5}$$

$$\bar{U}(\alpha') = -P_{\text{odd}}(\alpha) U_0 + P_{\text{even}}(\alpha) U_0.\tag{R6}$$

FIG. R2. Numerically evaluated norm of the directional displacement \mathbf{d} for 16-time-step quantum walks, with $\alpha \in [0, 1]$. The parameters are the same as those in Fig. 3e of the main text. The gray line marks $\alpha = 0.5$.

Under the design of the shift operator S_x , U_0 would transform state components on lattice sites with odd- x label to a superposition of its neighboring sites with even- x label, and vice versa. Hence, given the same initial states, the time-evolved states under $\bar{U}(\alpha)$ and $\bar{U}(\alpha')$ would differ only by local signs, which do not change the probability distribution over the lattice sites. For instance, in the experimentally relevant case of even number of time steps t , the time-evolved states under $\bar{U}(\alpha)$ and $\bar{U}(\alpha')$ would only differ by a global phase $(-1)^{\frac{t}{2}}$, which does not change the occupation distribution.

In the revised manuscript, we added detailed analysis on the dynamics with $\alpha \in [0.5, 1]$ in the Supplemental Material, and added a brief discussion in the main text.

7- I recommend not using “quantum” as there is nothing in the paper about quantum. The experiment is a random walker in a discrete lattice. Thus, it is misleading to use the term quantum.

We thank the Referee for the insightful comment, and we fully understand the Referee’s concern. Indeed, as we use attenuated laser pulses, the experiment is performed in the classical regime. Based on the coherence of the laser pulse, we are able to observe the same dynamic behavior as that of single photons. Throughout our manuscript, the term

“quantum” is almost always used in the combination “quantum walk”, which we believe has important differences with the classical random walk.

First, in a classical random walk, the time evolution of the walker is intrinsically stochastic, which is the origin of its spatial probability distribution at final times. By contrast, in quantum walks, as is the case with our experiment, the time evolution is deterministic, governed by the time-evolution operator U . At any instant of the evolution, the system is in a coherent superposition: a superposition of localized quantum states in the quantum regime (as in single photons for instance); but a superposition of different modes in the case of coherent light (as in our experiment). The observed intensity distribution of laser comes from the interference effect, with the electric-field vectors of the laser modes obeying the same linear-algebraic description as that of single photons (or quantum states in general). As such, we are essentially simulating the quantum interference using the coherence of laser.

Second, a classical random walk features distinct probabilistic distribution from that of a quantum walk. The spreading of the spatial distribution following a classical random walk is diffusive, with $\Delta x^2 \sim t$, whereas the spreading is ballistic $\Delta x^2 \sim t^2$ for quantum walks [c.f. Phys. Rev. Lett. **91**, 130602 (2003); Quant. Inf. Process. **11**, 1015-1106 (2012); Contemp. Phys. **44**, 307-327 (2003)]. It is only when the coherence (either of coherent light or quantum state) is destroyed that one may recover the diffusive dynamics from a quantum-walk setup.

Furthermore, the terminology “quantum walk” has been extensively used and widely accepted in the literature for systems similar to ours. We list some of the works below:

- Schreiber, A. et al. Photons walking the line: a quantum walk with adjustable coin operations. Phys. Rev. Lett. **104**, 050502 (2010).
- Schreiber, A. et al. A 2D quantum walk simulation of two-particle dynamics. Science **336**, 55-58 (2012).
- Chen, C. et al. Observation of topologically protected edge states in a photonic two-dimensional quantum walk. Phys. Rev. Lett. **121**, 100502 (2018).
- Weidemann, S., Kremer, M., Longhi, S. & Szameit, A. Topological triple phase transition in non-Hermitian Floquet quasicrystals. Nature **601**, 354-359 (2022).

Note that the last article above introduces a classical amplifier as gain, which is only feasible for classical light. This also demonstrates the advantage of our experimental setup, where the coherent light source can be directly replaced by a quantum source (for instance, heralded single photons created via spontaneous parametric down-conversion). However, the brightness of the state-of-the-art single photon source is not sufficient for our requirements, i.e., after a limited time of evolution, the experimental data will be dominated by noise. Nevertheless, if this issue can be solved, the mechanism underlying our setup may stimulate novel quantum-device design in the future.

In summary, the abstract needs a lot of work as it is not clearly written. The results are presented nicely but there is a significant lack of analytical and quantitative explanations for the observations. There are also technical terms that need to be used/removed as they provide the wrong message.

We thank Referee A for the constructive comments and suggestions, which give us the opportunity to revise and improve our work. We trust that this new iteration of our manuscript is suitable for further consideration in Nature Communications.

Reply to Report of Referee B

We thank Referee B for her/his careful reading of our manuscript and for recommending our work for publication in Nature Communications. Following the Referee's insightful comments and helpful suggestions, we have revised the manuscript accordingly. Below let us address the Referee's questions point-by-point.

This work reports the realization of non-Hermitian synthetic lattices supporting NHSE in quantum-walk experiments and, by adding effective (synthetic) magnetic flux, the influence of the flux on NHSE is experimentally explored. The subject involving quantum walk, effective magnetic flux, and also NHSE certainly attracts broad interest in both optic and quantum communities. The experimental results are clear to support the conclusions, especially the magnetic suppression of NHSE is of great importance for future flexible control of light (photons) in higher dimensions. I think this study should be published in high impact journal such as Nat. Commun. Here I list some comments for the authors to consider:

We thank the Referee for the constructive comments, and for recommending our work for publication.

1. It should be noticed (as far as I understand), Ref. [43] may have different mechanism as that in the current manuscript to build the synthetic lattice. The key difference is that delay lines in Ref. [43] guide only a small portion of the energy in each pulse so the experimental platform exactly builds a tight-binding lattice. However, in the current work, the platform supports two-dimensional photonic quantum walk. Firstly, I think this difference should be acknowledged. Secondly, although I can understand how such 2D quantum walk works, it is not very clear to me how it can be understood in a tight-binding model as shown in Fig. 1a. Further clarification is appreciated.

We thank the Referee for the insightful comment. While both experiments exploit the time-multiplexed scheme, there are indeed some important differences.

First, in Ref. [43], the decay lines introduce dissipative couplings between pulses (or synthetic sites), such that the dynamics is modeled by a Lindblad equation. By contrast,

in our setup, the coupling is implemented by the shift operators $S_{x,y}$, which, being unitary, mimic the time-evolution operators governed by polarization-dependent, coherent hopping terms. The non-Hermiticity in our experiment is introduced through the loss operators $M_{x,y}$.

Second, in our setup, we implement, for each cycle in the fiber loop, the non-unitary operator U . The effective Hamiltonian can be defined as $U = e^{-iH_{\text{eff}}}$, such that the quantum walk constitutes a stroboscopic simulation of the time evolution governed by H_{eff} . Formally, we can derive H_{eff} from its definition, but its form is complicated and does not provide much insight in the form of a tight-binding model.

As a compromise, we can work in the high-frequency limit, assuming each time step to be short enough that the time evolution can be Trotterized and kept to the lowest order. More specifically, we define the effective Hamiltonians for the components of the Floquet operator $U = M_y S_y P C M_x S_x C$, with

$$C = e^{-iH_1}, \quad M_x S_x = e^{-iH_2}, \quad M_y S_y P = e^{-iH_3}. \quad (\text{R7})$$

The overall time evolution within each time step is then approximately driven by the effective Hamiltonian

$$H'_{\text{eff}} = \frac{1}{4}(2H_1 + H_2 + H_3), \quad (\text{R8})$$

where

$$H_1 = \frac{\pi}{2\sqrt{2}} \begin{pmatrix} 1 - \sqrt{2} & 1 \\ 1 & -(1 + \sqrt{2}) \end{pmatrix} \otimes \mathbf{1}_r, \quad (\text{R9})$$

$$H_2 = \sum_{x,y} \left[e^{\gamma_x} \begin{pmatrix} \frac{i}{2} & 0 \\ 0 & -\frac{i}{2} \end{pmatrix} \otimes |x-1, y\rangle\langle x, y| - e^{-\gamma_x} \begin{pmatrix} \frac{i}{2} & 0 \\ 0 & -\frac{i}{2} \end{pmatrix} \otimes |x, y\rangle\langle x-1, y| \right], \quad (\text{R10})$$

$$H_3 = \sum_{x,y} \left[e^{i2\pi\alpha x} e^{\gamma_y} \begin{pmatrix} \frac{i}{2} & 0 \\ 0 & -\frac{i}{2} \end{pmatrix} \otimes |x, y-1\rangle\langle x, y| - e^{-i2\pi\alpha x} e^{-\gamma_y} \begin{pmatrix} \frac{i}{2} & 0 \\ 0 & -\frac{i}{2} \end{pmatrix} \otimes |x, y\rangle\langle x, y-1| \right]. \quad (\text{R11})$$

Apparently, while H_2 describes asymmetric hopping along the x direction, H_3 indicates asymmetric hopping along the y direction, with the Peierls phase on top. Thus, although $H'_{\text{eff}} \neq H_{\text{eff}}$, it offers a more transparent understanding of the quantum-walk dynamics from the perspective of a tight-binding model.

FIG. R3. Numerically evaluated norm of the directional displacement \mathbf{d} for 16-time-step quantum walks, with $\alpha \in [0, 1]$. The parameters are the same as those in Fig. 3e of the main text. The gray line marks $\alpha = 0.5$.

In the revised Methods, we derive the effective Hamiltonian above, and discuss our setup in the light of the corresponding tight-binding model. We also discuss the key difference from Ref. [43] in the main text.

2. In Fig. 3e, I see symmetric patterns with alpha from 0 to 0.5. Is there any explanation, as the flux is $2\pi\alpha$ and one may expect to see symmetric patterns with alpha from 0 to 1.

We thank the Referee for the helpful question. We chose the parameter range $\alpha \in [0, 0.5)$ because the dynamics of on-site occupation under $U(\alpha)$ and $U(\alpha + 0.5)$ are the same. To explicitly demonstrate this, we extend the parameter range of Fig. 3e in the main text, and show the numerically evaluated $|\mathbf{d}|$ in Fig. R3, with the observation that $|\mathbf{d}(\alpha)| = |\mathbf{d}(\alpha + 0.5)|$.

To understand this observation, we fit the Floquet operator U in a different time frame, i.e., subject it to a unitary transformation $\bar{U} = S_y^\dagger U S_y$, where S_y is the shift operator defined in the manuscript. Denoting $U_0 = M_y C M_x S_x C$, we have

$$\begin{aligned} \bar{U} &= P U_0 \\ &= P_{\text{odd}} U_0 + P_{\text{even}} U_0, \end{aligned} \tag{R12}$$

where

$$P_{\text{odd (even)}} = \begin{cases} \sum_{x,y} \begin{pmatrix} e^{i2\pi\alpha x} & 0 \\ 0 & e^{-i2\pi\alpha x} \end{pmatrix} \otimes |x,y\rangle\langle x,y|, & x \text{ is odd (even)}; \\ 0, & \text{otherwise.} \end{cases} \quad (\text{R13})$$

Consider $\alpha' = \alpha + 0.5$, we have

$$\bar{U}(\alpha) = P_{\text{odd}}(\alpha)U_0 + P_{\text{even}}(\alpha)U_0, \quad (\text{R14})$$

$$\bar{U}(\alpha') = -P_{\text{odd}}(\alpha)U_0 + P_{\text{even}}(\alpha)U_0. \quad (\text{R15})$$

Under the design of the shift operator S_x , U_0 would transform state components on lattice sites with odd- x label to those with even- x label, and vice versa. Hence, given the same initial states, the time-evolved states under $\bar{U}(\alpha)$ and $\bar{U}(\alpha')$ would differ only by local signs, which do not change the on-site occupation distribution. For instance, in the experimentally relevant case of even number of time steps t , the time-evolved states under $\bar{U}(\alpha)$ and $\bar{U}(\alpha')$ would only differ by a global phase $(-1)^{\frac{t}{2}}$, which does not change the occupation distribution.

In the revised manuscript, we added a detailed analysis of the dynamics with $\alpha \in [0.5, 1]$ in the Supplemental Material, and added a brief discussion in the main text.

3. The authors envision that their experiment can help study NHSE in higher dimensions. However, to the best of my knowledge, the current experimental platform is limited in the total number of time bins. If three dimensional lattice is to be studied, then the number of time bins is largely increased. Can the authors provide some perspective on this matter? Or in other words, will the researchers go to three dimensions in the near future?

We thank the Referee for the comment. The Referee is correct that a major limitation on our experimental setup is the requirement of a large number of time bins for simulating lattice dynamics in higher dimensions. However, as we demonstrate below, it is already practical to implement quantum walks in three dimensions using our setup.

The basic idea here is to devise a third-time scale Δ_z to encode the spatial dimension z , in addition to the existing time scales ($\Delta_x = 80$ ns, $\Delta_y = 4.83$ ns). For this purpose, we can use the birefringent crystals in a collinear cut to realize the shift operator S_z along the z direction. The birefringent crystals can create a time delay of ~ 10 ps between the horizontal

and vertical polarizations, which enables us to encode the positions in the z direction. A straightforward estimation ($4.83 \text{ ns}/10 \text{ ps} = 483$) shows that such a choice allows us to encode sufficient spatial positions in the z direction. Furthermore, the birefringent crystals, unlike the unequal-length interferometer that we use, are very stable with a high transmission rate. They do not reduce the overall stability of the experimental setup. Since modern detectors can resolve time differences of the order $\sim 1 \text{ ps}$, data detection is in principle not an issue, but may be limited by measurement noise.

As such, we believe it is possible for researchers to study quantum walks in three dimensions in the near future, with the number of time steps approaching 20 (the primary restriction here is the effectiveness of the optical loop).

4. Although I am usually fine with the phrase ‘quantum walk’, I do not see much ‘quantum’ here. The experiment is conducted with classical light (if not, please correct me). Such phenomena can also be named as ‘random walk’. Probably it will be good to discuss why this platform is ‘quantum’ and can be used for quantum device design.

We thank the Referee for the insightful comment. Indeed, as we use attenuated laser pulses, the experiment is performed in the classical regime. Based on the coherence of the laser pulse, we are able to observe the same dynamic behavior as that of single photons. Throughout our manuscript, the term “quantum” is almost always used in the combination “quantum walk”, which we believe has important differences with the classical random walk.

First, in a classical random walk, the time evolution of the walker is intrinsically stochastic, which is the origin of its spatial probability distribution at final times. By contrast, in quantum walks, as is the case with our experiment, the time evolution is deterministic, governed by the time-evolution operator U . At any instant of the evolution, the system is in a coherent superposition: a superposition of localized quantum states in the quantum regime (as in single photons for instance); but a superposition of different modes in the case of coherent light (as in our experiment). The observed intensity distribution of laser comes from the interference effect, with the electric-field vectors of the laser modes obeying the same linear-algebraic description as that of single photons (or quantum states in general). As such, we are essentially simulating the quantum interference using the coherence of laser.

Second, a classical random walk features distinct probabilistic distribution from that of a

quantum walk. The spreading of the spatial distribution following a classical random walk is diffusive, with $\Delta x^2 \sim t$, whereas the spreading is ballistic $\Delta x^2 \sim t^2$ for quantum walks [c.f. Phys. Rev. Lett. **91**, 130602 (2003); Quant. Inf. Process. **11**, 1015-1106 (2012); Contemp. Phys. **44**, 307-327 (2003)]. It is only when the coherence (either of coherent light or quantum state) is destroyed that one may recover the diffusive dynamics from a quantum-walk setup.

Furthermore, the terminology “quantum walk” has been extensively used and widely accepted in the literature for systems similar to ours. We list some of the works below:

- Schreiber, A. et al. Photons walking the line: a quantum walk with adjustable coin operations. Phys. Rev. Lett. **104**, 050502 (2010).
- Schreiber, A. et al. A 2D quantum walk simulation of two-particle dynamics. Science **336**, 55-58 (2012).
- Chen, C. et al. Observation of topologically protected edge states in a photonic two-dimensional quantum walk. Phys. Rev. Lett. **121**, 100502 (2018).
- Weidemann, S., Kremer, M., Longhi, S. & Szameit, A. Topological triple phase transition in non-Hermitian Floquet quasicrystals. Nature **601**, 354-359 (2022).

Note that the last article above introduces a classical amplifier as gain, which is only feasible for classical light. This also demonstrates the advantage of our experimental setup, where the coherent light source can be directly replaced by a quantum source (for instance, heralded single photons created via spontaneous parametric down-conversion). However, the brightness of the state-of-the-art single photon source is not sufficient for our requirements, i.e., after a limited time of evolution, the experimental data will be dominated by noise. Nevertheless, if this issue can be solved, the mechanism underlying our setup may stimulate novel quantum-device design in the future.

List of changes

1. We changed the title to “Manipulating directional flow in a two-dimensional magnetic quantum walk”.
2. We replaced the fourth sentence in the abstract by “The directional flow hereof underlies the emergence of the non-Hermitian skin effect, with its orientation continuously adjustable through the photon-loss parameters”.
3. We replaced the first sentence in the first paragraph by “Open systems are ubiquitous in nature, and exhibit rich and complex behaviors unknown to their closed counterparts”.
4. We replaced the fourth sentence of the second paragraph from by “In particular, when the directional flow is tuned to the diagonal of the square lattice, the system exhibits the much discussed corner skin effect [22]”. Reference [22] was also added here.
5. We added the discussion on the key difference from Ref. [43] in the revised main text.
6. We removed the terms “non-reciprocal” and “non-reciprocity” throughout the text, and revised the title and abstract accordingly.
7. We provided the detailed supporting analysis from the perspective of an effective tight-binding model in the revised Methods.
8. We added more discussions on the origin of the competition between the non-Hermitian skin effect and the magnetic confinement both in the revised main text and in the revised Supplemental Material.
9. We added numerical calculations of the eigenspectra and eigenstates distribution in the revised Supplemental Material and plotted the quasienergy spectra of the effective Hamiltonian H_{eff} under different magnetic fluxes in Fig. S5.
10. We added more discussions on the origin of the competition between the non-Hermitian skin effect and the magnetic confinement both in the revised main text and in the revised Supplemental Material.

11. We added a detailed dynamic analysis of the time-evolved states with $\alpha \in [0.5, 1]$ in the revised Supplemental Material (including Fig. S6), and added a brief discussion in the revised main text.

REVIEWERS' COMMENTS

Reviewer #2 (Remarks to the Author):

I thank the authors gave very detailed responses to all my previous comments. I find them are very helpful in addressing all my concerns and the manuscript, after revision, is also greatly improved. I believe this manuscript can be considered for publication in NC and would like to recommend the acceptance.

Reviewer #3 (Remarks to the Author):

I was asked to assess the reply of the authors to Reviewer A. Overall, the responses are convincing, and the authors implemented relevant changes according to the referee's comments and suggestions. These changes improve the readability and accessibility of their results, which are technically well presented.

In response to the criticism of the referee in point 7 on employing "quantum" for random walkers, the authors provided an extensive reply and shared relevant references. This point is also raised by the second referee and is very probable to come to the mind of the prospective readers of this work. I hence recommend the authors keep a concise summary of their response to the referee as a footnote (or the text) in the main text of the manuscript to eliminate future concerns in this regard.

Aside from the above-mentioned point, the manuscript reaches the standard to be published in Nature Communications.

Reply to Report of Reviewer 2

I thank the authors gave very detailed responses to all my previous comments. I find them are very helpful in addressing all my concerns and the manuscript, after revision, is also greatly improved. I believe this manuscript can be considered for publication in NC and would like to recommend the acceptance.

We are pleased that Reviewer 2 is satisfied with the revised manuscript, and thank the Reviewer for recommending our work for publication.

Reply to Report of Reviewer 3

I was asked to assess the reply of the authors to Reviewer A. Overall, the responses are convincing, and the authors implemented relevant changes according to the referee's comments and suggestions. These changes improve the readability and accessibility of their results, which are technically well presented.

We thank the Reviewer for his/her valuable time and found our responses convincing.

In response to the criticism of the referee in point 7 on employing quantum for random walkers, the authors provided an extensive reply and shared relevant references. This point is also raised by the second referee and is very probable to come to the mind of the prospective readers of this work. I hence recommend the authors keep a concise summary of their response to the referee as a footnote (or the text) in the main text of the manuscript to eliminate future concerns in this regard.

Aside from the above-mentioned point, the manuscript reaches the standard to be published in Nature Communications.

We thank the Reviewer for the comment. In the latest version of our manuscript, we added new discussions in the main text, clarifying the difference between quantum walk and classical random walk.